# Pandemic Preparedness among Big Energy Companies: Call to Research and Action

**DOI:** 10.3390/ijerph20042771

**Published:** 2023-02-04

**Authors:** Maria Rosaria Gualano, Leonardo Villani, Walter Ricciardi

**Affiliations:** 1School of Medicine, UniCamillus-Saint Camillus International University of Health Sciences, 00131 Rome, Italy; 2Section of Hygiene, University Department of Life Sciences and Public Health, Università Cattolica del Sacro Cuore, Largo Francesco Vito 1, 00168 Rome, Italy

**Keywords:** COVID-19, preparedness, occupational health, energy sector

## Abstract

The COVID-19 pandemic, as a global phenomenon, has affected all the working realities, worldwide, with the same issues. The aim of the present work is to assess the experiences of management and their preparedness during the pandemic among big companies, in particular, in the energy sector. Based on an overview of scientific evidence and grey literature, we found that big companies followed evidence-based decision-making practices and offered preparedness and information plans. Specifically, these plans contained recommendations and best practices to be followed to avoid the risk of infection in the workplaces, as well as in the field of epidemiological surveillance and vaccination. Nevertheless, many research efforts are required, and it is important that a large number of big companies and corporations address these challenges worldwide, adopting a new sustainable approach that includes both the productivity and health of the workers. A Call to Action was then issued in order to achieve evidence-based leadership to address current and future public health emergency scenarios.

## 1. Introduction

The COVID-19 pandemic showed that preparedness, management and recovery in the context of public health emergencies is critical for limiting the health, social, environmental and economic impacts. Such mechanisms should be adopted at the macro (governmental, national health service, international organizations), meso (regional, community) and micro (hospitals, local health authorities, companies) levels. In this context, it is crucial for companies to limit the impact of health emergencies by ensuring both the health and productivity of workers adopting sustainable actions. To date, however, there is limited knowledge about the best strategies and the best practices adopted by companies to reduce the impact of health emergencies, such as COVID-19. In this context, the objective of this study is to assess the experiences of management and their preparedness for the pandemic among big companies, through a review of the literature and the example of the actions adopted by a large Italian company. Therefore, it is possible to focus on the best practices and develop seven proposals needed to strengthen collaboration among businesses, governments, workers and citizens in managing health emergencies.

## 2. Background

Public health emergency preparedness is defined as “the ability of health and public health systems, as well as communities and individuals, to prevent, protect against, respond rapidly to, and recover from public health emergencies, particularly those whose magnitude, timing, or unpredictability threaten to unduly strain normal routine functions. “Preparedness” for public health emergencies involves a coordinated and continuous process of planning and implementation that relies on measuring performance and taking corrective action” [1].

The COVID-19 pandemic is a major stress test for health systems and governments around the world, with huge direct and indirect health, environmental, social and economic impacts [2,3,4]. Indeed, the pandemic represented a challenge for health systems, highlighting their strengths and weaknesses, both in terms of preparedness and the organizational and managerial responses [5].

At the same time, the pandemic acted as an accelerator of processes such as digitalization, telehealth, the development of new and innovative working models [6,7,8,9], with a focus not only on productivity, but also on the wellbeing of individuals and the environment, particularly from a Planetary Health perspective. Recently, the Rockefeller Foundation–Lancet Commission on Planetary Health adopted an operational definition of Planetary Health, developed by a panel of experts in all stakeholder fields, indicating it as “the achievement of the highest attainable standard of health, wellbeing, an equity worldwide through judicious attention to the human systems—political, economic, and social—that shape the future of humanity and the Earth’s natural systems that define the safe environmental limits within which humanity can flourish. Put simply, Planetary Health is the health of human civilization and the state of the natural systems on which it depends” [10].

This approach can be applied to different contexts, and it is based on shared and effective governance, communication, collaboration and coordination. At the same time, rethinking work, education, research, and communication activities within this new approach can help foster the implementation of policies and equitable and holistic solutions that take into consideration the need to protect both human and planetary health. In this context, the need to integrate the Planetary Health approach into preparedness systems to improve the prevention of, and response to, global threats and promote sustainable development is evident [11,12,13].

For instance, as a response to the COVID-19 pandemic and to better deal with future emergencies, the European Union (EU) established the Health Emergency Preparedness and Response Authority (HERA) in order to prevent, detect and respond rapidly to health emergencies. Its main purpose is to face threats and potential health emergencies through strengthening the connections and networks among member countries, and ensuring the adequate production and distribution of medicines, vaccines and other medical countermeasures [14].

As has happened at the global and community level, even at the micro level, i.e., of individual corporates, companies and enterprises, the pandemic and the worsening of the climate crisis have led to a change in the management and implementation of working activities, favoring a vision more directed toward Planetary Health approaches.

In this context, the organizational and managerial system regarding both the health protection of workers and the continuity and productivity of business activities was oriented through innovative and resilient models, based on the use of the new tools and technologies. Therefore, preparedness and the ability to respond to major emergencies has become a priority for companies [15].

In the current scenario, the social and environmental commitment, along with good governance practices, have even greater importance. In this context, the criteria of “Environmental, Social, Corporate Governance (ESG)” were introduced in 2004, in order to define the type of investment in in terms of financial, ethical, environmental and social impact [16]. Given the increasing focus on climate change, many companies have joined the ESG system to ensure accountability and impact management systems with positive results [17]. Indeed, the data from a recent report demonstrate that companies with a proactive ESG strategy performed better on both COVID-19 and inequality in the period from the onset of the COVID-19 crisis [18]. Sustainability, therefore, is not only defined from the financial and economical perspective, but also including environmental and social aspects.

In this context, in November 2021, the United Nations (UN) Climate Change Conference 2021 (Conference of the Parties—COP26) strengthened the need to contain the rise of the global temperature [19], while the EU has enshrined a commitment to achieve carbon neutrality by 2050 [20]. The aim is to realize the energy transition through the realization of a shift from energy sources centered primarily on fossil fuels to cleaner, renewable sources with very low or zero carbon emissions. A key contribution to these processes comes from the digitalization of power grids and systems, which now impacts the production and community sector, and could also ensure better energy efficiency and more sustainable development. In addition, the Russia-Ukraine conflict in the heart of Europe, which began in February 2022, is causing repercussions on food and energy supply sources. The International Energy Agency is currently monitoring the implications of the geopolitical crisis on energy markets [21].

Thus, it seems essential to have reliable data available in order to propose and initiate actions to best address the current and future challenges of a large enterprise in an evidence-based (i.e., science-based) approach.

In this scenario, companies must deal with many great challenges that are interconnected, having to consider challenges related to climate change, health emergencies (related to infectious diseases) and, at the same time, the wellbeing of workers. Furthermore, in addition to emergency preparedness and management, it is paramount to consider issues that arise as a result of health crises. One example is the COVID-19 pandemic, which impacted the health of the population and workers by leaving post-infection sequelae (long-COVID) [22], that could affect not only individual’s health status, but also productivity.

It is therefore important that companies implement specific programs for emergency preparedness, management and recovery, taking into consideration the health of the planet as well as the workers, implementing actions that are sustainable and effective.

In this context, the purpose of this perspective paper is to describe the global scenario of the best practices of COVID-19 preparedness and response plans among energy companies in order to quantify and detail the contribution that companies can make in defining actions and best practices and to be able to release a call to action.

## 3. Analyses of Scenario

In order to identify the best practices adopted by energy and non-energy companies worldwide during the COVID-19 pandemic, in terms of preparedness and management, we conducted a literature review with a double approach (scientific database and grey literature searches).

Firstly, a search of the relevant articles was performed on PubMed, without time and language limitations. The last update was on 3 December 2022.

The keywords and search strings were produced through Boolean operators’ combination, by using the following terms: (“energy corporate” AND COVID * AND “pandemic preparedness” AND (corporate OR company *)); (preparedness AND COVID AND corporate); (preparedness AND COVID AND “energy company *”).

The selection of the articles of interest was carried out following the chosen eligibility criteria, related to the descriptions of pandemic preparedness and management practices and action implemented by energy and non-energy companies during the pandemic.

In addition, we performed a grey literature search through the generalist search engine Google, which is one of the top four sources of online information in the world [23,24]. In this case, the research focused on actions taken as preparedness and response to pandemic waves in the contexts of global energy companies, with the following search query: (“energy corporate” AND COVID AND management).

The search and subsequent selection of the sites of interest was conducted considering the presence within the website of descriptions of pandemic preparedness and management practices by energy companies.

The flowchart in Figure 1, based on the PRISMA 2020 Statement [25], shows the inclusion process of the articles and reports. In order to assess the eligibility of articles, two researchers firstly conducted a screening for the title and abstract independently. The cases where there was no agreement were discussed and resolved. At the end of this phase, six papers proved to be congruent with the purpose of our research, thus including information on preparedness and responses plan to the pandemics. Then, we evaluated the eligibility of the include papers by full text. The final screening resulted in the inclusion of three papers [26,27,28], as discussed below.

The paper published by Maldin-Morgenthau and colleagues [26] describes the results of a roundtable discussion held in 2006 in New York City, attended by 20 major USA corporate and business stakeholders from different industry sectors: information technology, finance, pharmaceuticals, transportation, aerospace, consumer products, utilities, communications, energy and defense. The participants included 21 individuals from 20 companies with responsibilities for business continuity, corporate security, emergency preparedness and response and risk management. The roundtable offered executives from leading local and international companies an opportunity to share information and discuss the challenges they face in preparing their organizations, customers, suppliers and employees for a possible pandemic. The meeting featured presentations from the Center for Biosecurity, The Bellwether Group, Inc. and the New York City Department of Health and Mental Hygiene, followed by a structured discussion among the participants. Prior to the panel discussion, a survey was submitted in order to assess the status of corporate planning against pandemics. Indeed, the objectives were to discuss the current threats of the pandemic influenza and the possible corporate response to it; to determine the state of preparedness within the participant group; to identify the best practices in corporate preparedness for pandemic influenza; and to provide an opportunity for the participating companies to learn from each other. The main concepts that emerged were related to many aspects of management, such as leadership skills and an alliance between all societal stakeholders, including governments and institutions; effective communication with employees and the general population; implementation of smart working systems; the capacity for system resilience and flexibility to adapt to the challenges presented; the ability to provide healthcare assistance to workers and their families, considering both physical and mental health. They concluded that business leaders have to collaborate, with preparedness plans released and implemented at the central government level for a pandemic and that “The business community can have an extremely positive impact on government actions, and there has never been a more important time to use their influence.”

The second paper [27] focused on preparedness during the COVID-19 in airline operations in a corporation in the USA. A questionnaire was administered to workers between May and June 2020. The primary outcome was understanding the perception of companies’ preparedness and response to the COVID-19 outbreak; the secondary questions were about risk management systems and the appropriateness of the response. The responses showed that airline risk management systems were perceived as a weakness in organizations; however, it was recognized that progressive steps were taken to improve their risk management protocols as a response to the pandemic. More proactive systems and priori pandemic impact assessments were recommended by many responders.

Finally, the third study [28] demonstrated that the strategy to regularly test asymptomatic workers is a highly effective measure to reduce transmission of SARS-CoV-2 in workplaces.

The second part of our search was focused on the grey literature and resulted in 1,040,000 entries. Analyzing the first ten pages, the first 100 websites were explored and analyzed as a sample. Of these, 39 directly linked or were directly linkable to the sites of energy industries. Of these, however, only 21 contained information on protocols and actions carried out by the company either as preparedness or to deal with the COVID-19 pandemic (eligibility criteria for inclusion). After the screening, five websites had a section specifically devoted to information on COVID-19. Considering the geographical distribution, North America and Europe account for the largest share of the information, dealing with many domains of preparedness and the management of the pandemic (Table 1).

Some companies report on their websites that, implementing all measures according to the Public Health Agencies, they have had zero COVID-19 cases attributable to transmission that occurred in the workplace. This evidence-based approach appears to be very important, not only at the corporate level, but also at the level of the decisions made by central governments, around the world, for the entire community. Indeed, the development of management approaches based on the evidence available in the current scientific literature and taken up by the world’s leading Public Health Institutions and Agencies is desirable to also guide decision making processes in the context of managing situations such as pandemic crises.

Finally, the retrieved websites that were considered eligible for inclusion in our analysis were analyzed in detail and, after a screening of all the webpages, we summarized the best practices implemented. In this context, we identified nine actions as best practices (i.e., a community intervention with potential great benefits and positive impacts, as demonstrated and shared by the scientific community, as vaccination campaigns). Table 2 shows and summarizes the main examples of the best practices described in the websites as result of the review.

## 4. The Italian Context: The Example of ENI Group

Among our findings, at the European level, an example of a best practice is the one undertaken by Eni group. After the beginning of the COVID-19 pandemic in February 2020, a protocol between the Italian Government, Confindustria (the main association representing manufacturing and service companies in Italy, with a voluntary membership of more than 150,000 companies) and labor organizations was signed on March 14 [29]. This protocol contains the organizational and management measures that are indispensable to continue to guarantee business and productivity while containing the risk of contagion, thus preserving the workers’ health. All Italian companies were required to adopt the protocol, adapting it to the context of their workplace.

In this context, Eni, an Italian energy company with more than 30,000 employees, operating in 69 countries worldwide [30], adopted a “Health, Safety, Environment, Security and Public Inconvenience” risk management model by resolution of the Board of Directors [31]. A Crisis Unit and COVID-19 Risk Management Committees were established within the companies and continuous updates about the preventive measures were provided to workers. At the governance level, Eni’s health function has an in-house health emergency management competence center, which constantly monitors the worldwide epidemiological data and supports business units through several actions, such as timely epidemiological updates about guidelines released by national and international institutions; the best practices in clinical management; information about immunization and travel medicine recommendations; and support in defining technical specifications for related services.

Moreover, the emergency management competence center published procedures, documents and guidance based on the directives of the national and international health authorities (Ministry of Health, WHO, CDC, European Centre of Disease Control—ECDC -) and industry associations (such as the International Association of Oil and Gas Producers—IOGP, International Petroleum Industry Environmental Conservation Association—IPIECA, and Oil and Gas UK—OGUK).

In particular, these procedures included the activation of Pandemic Preparedness Response Plans (PPRPs) and the integration into the Medical Emergency Response Plan (MERP) of all Eni employer lines and companies, with operational and management specificities to deal with the emergency. These documents deal with various activities and pathways to manage pandemic-related events, such as coordination between the Eni Crisis Unit and workers, the implementation of preventive and restrictive measures, the management of epidemiological surveillance and quarantine, the identification of priorities for garrisoning production activities, the relief of expatriate and family personnel and a review of the rotation and shift change arrangements. In addition, prevention and management measures were applied in the workplace, as reported in Table 3.

In this context, the Crisis Unit had a central role, acting as a specialized support and high-level advisor to the third-level central institutions, indicating the strategic guidelines for the transversal management of the health emergency, and defining the technical and organizational measures to be implemented.

Considering the health prevention and surveillance, mainly dedicated to the people in the workplace, Eni adopted a strategy based on various actions aimed at reducing travel, crowding of workplaces and outpatients’ ambulatories, and encouraging the use of personal protective equipment (PPE). For example, it was made mandatory to reserve medical examinations by appointment, postponing non-urgent visits, and investigations that increased the risk of spreading the virus (such as spirometry) were eliminated.

Finally, in accordance with the government directives, the employer and the competent physician had to cooperate with the Local Health Authorities in the occurrence of positive cases in the company.

As the company has several locations abroad, the Crisis Unit has also dealt with the management of workers abroad; in particular, through constant communication (good practices, behaviors to be adopted to avoid infection) and the monitoring of both country conditions (risk scenario) and the health status of the workers, facilitating repatriation when appropriate.

Finally, all stakeholders in the productive world should deploy all of the available tools and resources to come up with win-win solutions to aid science. Interestingly, Eni has collaborated with the European project EXSCALATE4CoV (EXaSCale smArt pLatform Against paThogEns) [32] in order to identify the safest and most promising drugs in the fight against SARS-CoV-2, through the use of the HPC5 technology, a supercomputer that stands as the world’s highest performing industrial supercomputing infrastructure, capable of reaching a processing power of 51.7 Petaflop/s peak, which enables it to perform 70 million billion mathematical operations in one second.

In the European project, EXSCALATE4CoV, the common goal was to conduct molecular dynamics simulations of the proteins present on the surface of the virus, which play a key role in the virus’ infection mechanism. The application of these tools to medicine, and in particular in pharmacology, can help in the current scenario, and future scenarios, to discover new therapeutic opportunities.

## 5. The Road towards a Call to Research and Action

Investing in the health and wellbeing of workers is indispensable for all companies that want to make a substantial contribution to sustainable development in terms of the social, health and community health perspectives. Moreover, ensuring the health of workers means retaining optimal results of productivity and the achievement of the company’s objectives. Businesses can make an important contribution to improving and maintaining the wellbeing of workers, both in terms of physical and mental health, which is an increasing priority from a public health perspective [33,34].

Analyzing our results, there are some important elements that have to be addressed: the companies that achieved important goals during the pandemic applied strategies based on scientific evidence and the advice and guidelines released by public health institutions worldwide. The majority of the retrieved studies and information were about companies located in the USA.

It is important to highlight that the CDC [35] has developed several comprehensive frameworks and toolkits in order to support companies to plan effectively for pandemic strategies, based on three pillars: “Preparedness and Communication” (activities that should be undertaken before the pandemic to ensure preparedness, and the communication with all segments of society); “Surveillance and Detection Monitoring” (surveillance systems that provide continuous “situational awareness,” to give the earliest warning to protect the population); and “Response and Containment” (strategies implemented to limit the spread of the outbreak and to mitigate the health, social and economic impacts of the outbreak).

Moreover, an important contribution could be given by the introduction of Artificial Intelligence (AI) systems, which could be useful to develop early detection and predictive models, analyzing large databases to make comparisons and rapidly obtain fundamental information to improve surveillance and Horizon Scanning systems worldwide.

In addition, our findings highlight the importance of undertaking vaccination campaigns in the workplace, which could result in a high level of coverage and could add a significant value to the immunization of the whole population [36].

An important aspect that must be deeply investigated is the mental health of workers, which still represents an under-addressed impact [37,38].

In addition, the pandemic showed how important it is to strengthen the collaboration between different health professionals; for example, the occupational physician (OP) and general practitioner (GP) should work in synergy to ensure the high quality health care of workers, operating in a network perspective with private companies and institutions of each country [39]. A multidimensional system in which Public Health services, GPs and Ops are the pivots should be the common base to achieve a higher value of healthcare.

Finally, the present overview has some limitations that must be acknowledged. First, we did not perform a systematic review of the literature, and we did not explore scientific databases other than PubMed, but this was not the specific aim of our overview. Nevertheless, as we used the most commonly used scientific search engine, we retrieved a very poor level of literature on the topic, so it would be important to underline that scientific research should address the companies’ workplace and health status and their protection during health emergencies. Additionally, corporations should be more compliant to participate in investigations in order to promote research on this important issue and to share the data at an international level to provide comparisons between countries.

## 6. Conclusions

Even if further studies and investigation are needed, we can argue that, in the current scenario, there are several open challenges that will have an impact in the post-COVID-19 era, and consequently, an alliance between all of the key players in the systems of governance and leadership of the different productive sectors is strongly recommended.

A joint effort in which major private players can act together to ensure human and planetary health must be the basis for common planning, which, in accordance with international Institutions and Health International Regulations, can provide an impetus for economic and sustainable growth worldwide.

As also remarked by the recent report released by the Pan-European Commission on Health and Sustainable Development [40], in order to prevent a repeat of a catastrophe on the same scale as the COVID-19 pandemic, some actions are needed: a Planetary Health policy, centered on the recognition of the interconnection between human, animal and environmental health; the need, at the global level, to address health, social, economic and gender inequalities; the centrality of making investments in innovation, data science and new technologies.

In this context, investment in research and scientific progress will place those organizations that are farsighted enough to do so as global leaders, in a position to lead the processes of change and evolution that are taking place. In addition, the experience of the supercomputer could be very interesting for the future in the light of the development and improvement of AI systems. To be a leader in this scenario, big companies cannot avoid participating in and expressing their interest in the actions that national governments and international organizations are putting in place, such as the G20 and the recent document drafted in Rome on health [41], as well as the development of the new International Pandemic Treaty, adopted within the framework of the WHO [42].

Strengthening preparedness, including at the level of the organization of international mobility, pharmaceutical and essential supply chains, and the overall vision leading to zoonotic risk reduction and increased population compliance in times of health-social crises are aspects that, with the help of the new and very powerful technological tools, leaders of the different productive sectors should address.

In modern societies, the central role of the citizen/worker/patient is acknowledged, as we have also begun to define patient centered healthcare. Finally, information and the empowerment of people should be another important issue that both the public and private sector should consider and investigate.

## 7. The 7 Actions to Undertake—Call to Action

Considering the results of our work, the following proposal for actions (Table 4) aims at promoting better levels of preparedness and the management of health emergency situations within companies that have a role as major private players, involving all the stakeholders, from institutions to citizens’ associations.

## Figures and Tables

**Figure 1 ijerph-20-02771-f001:**
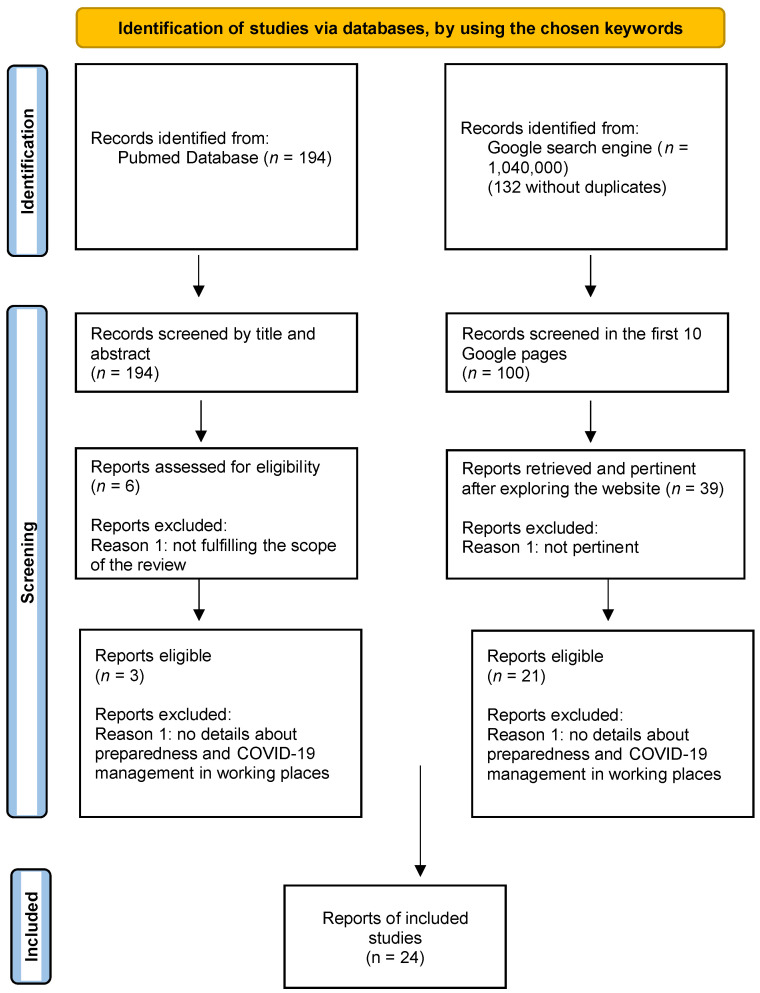
Flowchart showing the search strategy. Model of Flowchart modified from: [14,25].

**Table 1 ijerph-20-02771-t001:** Protocols and actions carried out by the companies during the COVID-19 pandemic.

Actions and Information Available for Workers and Corporate Policy Interventions
Activation of Pandemic Preparedness Plan
COVID-19 Incident Command System sessions
Engagement of the Corporate Response Plan Team
information and communication system of continuous updates with information from World Health Organization (WHO) and Centers for Disease Control (CDC)
Protocols developed following Occupational Safety and Health Administration (OSHA) guidelines
Working in concert with regulatory agencies, government entities, and emergency management organizations
Use of digital tools to inform and communicate
Adopting and strengthening hygiene measures and offering personal protective equipment (PPE) and testing
Implementation of smart working
Vaccinations
Participation in campaigns to donate devices and support materials
Investment in ventilation systems
Organizing informational meetings with employees

**Table 2 ijerph-20-02771-t002:** Best practices: strategies and examples.

Companies’ Sector	Best Practice for Employees	Setting	Topic	Source
Energetics	Pandemic Preparedness Plan	Texas (USA), Europe	COVID-19 management	Google
	Respiratory care program in COVID-19 times	Philippine	COVID-19 management	Google
	Mental Health Program	Malesia	COVID-19 management	Google
	TASK FORCE and COVID-19 Employer Survey Results	USA	COVID-19 management	Google
	Testing program	Canada	COVID-19 management	Google
	Comprehensive emergency response plans (ERPs)	USA	COVID-19 management	Google
	On-site vaccinations for employees and family members in 2021	USA, Canada, Europe	COVID-19 management	Google
Aeronautics			COVID-19 management	PubMed
Multisectoral	Roundtable for Pandemic Preparedness Plan		Flu Pandemics in general	PubMed

**Table 3 ijerph-20-02771-t003:** Prevention and management measures adopted by the Eni group to fight the pandemic in the workplace.

Activities Aimed at Workers to Reduce the Risk of Infection
Communication and information through posters and brochures, information sheet on entry at Eni sites for employees
Methods of access and control in workplaces (temperature detection at the entrance of sites, with prohibition of access if it is higher than 37.5°; mandatory signature and authorization for contractors and visitors, in accordance with the prevention and behavior measures
Personal workstations and cleaning/sanitation in all workplaces
Adoption of the smart working as the prevailing measure for carrying out work activities. Progressive mechanisms for alternating work and presence, with reference to office locations and compatible tasks
Adoption of specific personal protective equipment in relation to the types of activities performed

**Table 4 ijerph-20-02771-t004:** The seven actions to strength the collaboration between companies, governments, workers and citizens to manage health emergencies.

Call to Action to Strength Collaboration between Companies, Governments, Workers and Citizens
1. Strengthening Preparedness systems and drafting Emergency Preparedness Plans in a One Health perspective.
2. Strengthening Horizon Scanning systems for close monitoring of epidemiological emergencies, also and especially in collaboration with international Public Health Institutions and Agencies
3. Strengthening Use and provision of all technologies and digital platforms that can support public health medicine during emergency situations
4. Strengthening Support for the construction of Public Private Partnerships
5. Strengthening Implementation of evidence-based communication systems to keep citizens/workers informed and updated on prevention and health promotion issues
6. Strengthening Development of leadership systems at the corporate level that provide a vision always based on existing scientific evidence to support decision-making levels
7. Strengthening Willingness to build health surveillance models working together with the international Health Services, national and local institutions

## Data Availability

Not applicable.

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
