# Peer review of "Pandemic Preparedness among Big Energy Companies: Call to Research and Action"

_ijerph, 2023, doi:10.3390/ijerph20042771_

Round 1
Reviewer 1 Report (Previous Reviewer 2)
Please refer the attachment.

Author Response
Dear Reviewer, Thank you for your comments and suggestions. We have restructured the paper, focusing more on the concepts related to the objective of our work. Tables have also been added to facilitate better understanding of the text. Finally, the bibliography has been expanded and improved.
Reviewer 2 Report (New Reviewer)
The subject of the article is undeniably a part of the trend of journal. It contains elements of a novel character. It presents and organizes knowledge within the study area in a logical manner. In the article one can see the embedding of the issue in its proper context. The proportions of particular parts have generally been selected in a correct way. The layout of the paper is correct. The article closely corresponds to the topic specified in the title. The introduction to the topic is well prepared. The scenario shows the methodology procedure. The results of the analysis are adequate, but should be supported by the current literature. Review of the current international literature is one of the primary determinants of the quality of the article and is an important element in its evaluation. Hence, the number of used items of literature should be significantly increased, with particular emphasis on articles indexed in international databases.
Author Response
Dear Reviewer, Thank you for your comments and suggestions. We have tried to improve the paper by focusing more on concepts related to the objective of our paper. Tables have also been added to facilitate better understanding of the text. Finally, we have taken up the suggestion to expand the bibliography, which is now substantial and specific to the topic of our paper.
Reviewer 3 Report (New Reviewer)
Recommendations to the authors for improving their paper:
Section “Abstract”
Authors write: “The aim of the present work was to present a perspective ….”. The aim of scientific paper should not be “to present”. Is should be, for instance: evaluating, recognition of a phenomenon, etc.
The paper should start from section “Introduction”. The research problem, research gap, aim of research and research contribution should be provided in this section. There is also a lack of definition of research questions that are important in the case of a literature review.
Section “Background” should be after section “Introduction”. The current version of this section contain only facts and definitions. There is lack of analysis of other research related to topic of this research and lack of analysis of limitations of existing research. The references for related research should be extended.
There is lack of separation of section number 3. There is sub-section “3.3. The Italian context: the example of ENI group”, however I cannot find sub-sections 3.1 and 3.2.
General remark: The paper is poorly edited. For example, the layout of the text is non-uniform and the line spacing is non-uniform.
Author Response
Dear Reviewer,
Thank you for your comments and suggestions. We improved the article by: inserting tables to facilitate better understanding of the text; expanding the bibliography by adding references consistent with the article. Finally, we modified the layout of the article.
Round 2
Reviewer 1 Report (Previous Reviewer 2)
The authors already revised this paper according to the reviewer's comments.
Author Response
Dear Reviewer,
Thank you for the positive feedback. We have modified the paper in accordance with reviewer 3, adding a brief introduction outlining the purpose and objectives of the work. Finally, we have improved English language and style.
Reviewer 3 Report (New Reviewer)
The authors took into account most of the reviewer's comments.
The following comment should be also included:
The paper should start from section “Introduction”. The research problem, research gap, aim of research and research contribution should be provided in this section. Section “Background” should be after section “Introduction”.
Author Response
Dear Reviewer,
Thank you very much for the valuable suggestion, which we included by adding a short introduction explaining the research problem, research gap, aim of research and research contribution. In particular, we added the following part:
- Introduction
The COVID-19 pandemic showed that preparedness, management and recovery in the context of public health emergencies is critical to limiting health, social, environmental and economic impacts. Such mechanisms should be adopted at the macro (governmental, national health service, international organizations), meso (regional, community) and micro (hospitals, local health authorities, companies) levels. In this context, it is crucial for companies to limit the impact of health emergencies by ensuring both the health of workers and the productivity, adopting sustainable actions. To date, however, there is limited knowledge about the best strategies and the best-practices adopted by companies to reduce the impact of health emergencies, as well as COVID-19. In this context, the objective of this study is to assess the experiences of management and preparedness of the pandemic among the big companies, through a review of the literature and the example of the actions adopted by a large Italian company. Therefore, it is possible to focus on best practices and develop seven proposals needed to strengthen collaboration among businesses, governments, workers and citizens in managing health emergencies.
Finally, we improved English language and style.
Thanks for all the valuable suggestions that helped us improve our manuscript.
This manuscript is a resubmission of an earlier submission. The following is a list of the peer review reports and author responses from that submission.
Round 1
Reviewer 1 Report
The paper looks like a policy paper and not a "real" review paper.
The review is actually very poor - the reference list consists of 21 titles, and almost half of them belong to the "gray" literature.
I suggest an extension of the literature titles, especially from the scientific side. The Discussion section should be rewritten, with a critical analysis of the relevant findings.
The grey literature analysis is also very poor - from 14,200 entries, only 34 were directly linked to sites of energy industries. Of these, only 19 contained information on protocols and actions carried out by the company either as preparedness or to deal with the Covid-19 pandemic.
For the new extended list, I recommend the authors decide on criteria for information analysis, and to discuss the findings according to these criteria
The authors are insisting too much, in my opinion, on the case studies (ENI, and the European project EXSCALATE4CoV) with no very clear connection with the paper's objective.
The 7 Actions to undertake - Call to Action are very broadly described (Strengthening Support for the construction of Public Private Partnerships, for instance), and it is not clear if they really resulted from the study of the literature.
There are still mistakes in the text, for instance:
That enables it to perform 70 million 366 billion mathematical operations performed in one second
Author Response
The paper looks like a policy paper and not a "real" review paper.
We thank the reviewer for his valuable comments, we have actually the possibility to explain that the aim of the paper (that was totally revised) is to give an overview of the current knowledge on the topic (as in the title of our article) and the present work has not the ambition to perform a systematic review of the literature. However we performed a search in two different ways and we also want to highlight that more studies are needed to investigate and address the important challenge of preparedness in workplaces. The paper is submitted as a Perspective.
The review is actually very poor - the reference list consists of 21 titles, and almost half of them belong to the "gray" literature.
The paper was totally revised, the list of reference also is longer of course.
I suggest an extension of the literature titles, especially from the scientific side. The Discussion section should be rewritten, with a critical analysis of the relevant findings.
Thank you, we have revised the whole paper, also the discussion section contains more details, references, comparison, evaluation of strengthens and limitations of our findings.
The grey literature analysis is also very poor - from 14,200 entries, only 34 were directly linked to sites of energy industries. Of these, only 19 contained information on protocols and actions carried out by the company either as preparedness or to deal with the Covid-19 pandemic.
For the new extended list, I recommend the authors decide on criteria for information analysis, and to discuss the findings according to these criteria
We have acknowledged this in the revised sections of the manuscript.
The authors are insisting too much, in my opinion, on the case studies (ENI, and the European project EXSCALATE4CoV) with no very clear connection with the paper's objective.
Thanks for your suggestion, the paragraph are now more concise and connected to other revised sections of the paper.
The 7 Actions to undertake - Call to Action are very broadly described (Strengthening Support for the construction of Public Private Partnerships, for instance), and it is not clear if they really resulted from the study of the literature.
Thanks for your suggestion, we have acknowledged this in the revised version of the manuscript.
There are still mistakes in the text, for instance:
That enables it to perform 70 million 366 billion mathematical operations performed in one second
We have acknowledged this in the revised version of the manuscript.
Reviewer 2 Report
Please refer the attachment. Thanks!

Author Response
Thank you, we have revised the whole paper, also the results and discussion sections contain more details, references, comparison, evaluation of strengthens and limitations of our findings. We have of course updated the list of reference and language editing also, as requested.
We want to thank the reviewer for his valuable comments and would like to explain that the aim of the paper (that was totally revised) is to give an overview of the current knowledge on the topic (as in the title of our article) and the present work has not the ambition to perform a systematic review of the literature. However we performed a search in two different ways and we also want to highlight that more studies are needed to investigate and address the important challenge of preparedness in workplaces. The paper is submitted as a Perspective.
We think that, since very few papers exist in the scientific literature, our call to action and our findings should be a starting point important for future research that are strongly needed on this topic. Wellbeing of workers and their family is the wellbeing of the whole population but more data are needed on this and our perspective work could be of added value for this reason.
Round 2
Reviewer 1 Report
I recommend the acceptance of the paper, the manuscript was changed according to my suggestions
Reviewer 2 Report
LINE 71 andt is and
The English writing still needs to be much more improved.
LINE 410 The sentence is finished or not? (no period, and too short)
After reading thoroughly this article, I am not persuaded this article meets the standard and quality of SSCI/SCIE indexing Journal. However, the finial decision depends on your editor’s decion.